

# A proof-of-concept point-of-care test for the serodiagnosis of human amebic liver abscess

Rutchanee Rodpai[1,2], Penchom Janwan[1,3], Lakkhana Sadaow[1,4], Patcharaporn Boonroumkaew[1,4], Oranuch Sanpool[1,4], Tongjit Thanchomnang[1,5], Hiroshi Yamasaki[6], Toshihiro Mita[7], Pewpan M. Intapan[1] and Wanchai Maleewong[1,4]

[1] Mekong Health Science Research Institute, Khon Kaen University, Khon Kaen, Thailand
[2] Department of Medical Technology, Faculty of Allied Health Sciences, Nakhonratchasima College, Nakhon Ratchasima, Thailand
[3] Department of Medical Technology, School of Allied Health Sciences, Walailak University, Nakhon Si Thammarat, Thailand
[4] Department of Parasitology, Faculty of Medicine, Khon Kaen University, Khon Kaen, Thailand
[5] Faculty of Medicine, Mahasarakham University, Maha Sarakham, Thailand
[6] Department of Parasitology, National Institute of Infectious Diseases, Tokyo, Japan
[7] Department of Tropical Medicine and Parasitology, Juntendo University School of Medicine, Tokyo, Japan

Corresponding author
Wanchai Maleewong,
wanch_ma@kku.ac.th

## ABSTRACT

**Background.** Amebic liver abscess (ALA), caused by an extraintestinal invasion of the virulent protozoan *Entamoeba histolytica*, is important among parasitic causes of morbidity and mortality, especially in the tropics. Clinical symptoms, medical-imaging abnormalities of the liver and serological tests are normally made for supportive diagnosis. Serum-based enzyme-linked immunosorbent assay (ELISA) has been conventionally used for diagnosing ALA but is time-consuming and sophisticated equipment is required. Therefore, we sought to develop a new and rapid innovative point-of-care immunochromatographic test (ICT) that can use whole blood as an alternative to serum-based ELISA. An ICT tool using simulated whole-blood samples was developed for immunoglobulin G antibody detection, and its diagnostic efficiency was evaluated in comparison with serum-based ELISA.

**Methods.** Both methods were tested to assess their diagnostic performance using a total of 253 serum samples. These came from ALA patients ($n = 13$), healthy individuals ($n = 40$), and patients with other diseases ($n = 200$).

**Results.** Amebiasis-ICT exhibited 100% (95% confidential interval (CI) [75.3–100.0]) sensitivity and 97.1% (95% CI [94.1–98.8]) specificity, whereas ELISA gave the same sensitivity (100% 95% CI [75.3–100.0]) and slightly lower specificity (95.8% 95% CI [92.5–98.0]). There were no significant differences in sensitivity and specificity between the two tests (Exact McNemar's test; $p > 0.05$), with Cohen's kappa agreement 96.44% ($\kappa$-value $= 0.771$, $p < 0.001$) indicating substantial agreement.

**Conclusion.** This ICT tool using simulated whole-blood samples has a high possibility of being used with real whole blood. Therefore, since there is no need to separate serum, this can be considered an innovative diagnostic tool to replace serum-based ELISA in clinics and field surveys in remote areas where medical facilities are limited.

## INTRODUCTION

Amebiasis is caused by the protozoan *Entamoeba histolytica* and is one of the most important neglected tropical diseases. Amebiasis has higher morbidity and mortality than many intestinal parasitic protozoan infections, causing an estimated 40,000–110,000 deaths each year (*Lozano et al., 2012*; *Gupta, Smith & Diakiw, 2022*; *Usuda et al., 2022*; *Datta et al., 2024*). The highest infection rates have been reported in low-income countries, especially West and East Africa, Central and South America, and regions in South and Southeast Asia (*Stanley, 2003*; *Shirley et al., 2018*). Infection occurs most commonly through the fecal-oral route with the ingestion of water or food contaminated with cysts.

Amebic liver abscess (ALA) is a severe extraintestinal manifestation caused when pathogenic trophozoites of *E. histolytica* disseminate to the liver. Onset of ALA produces varied symptoms that may occur within a few weeks to several months or years (*Lachish, Wieder-Finesod & Schwartz, 2016*; *Gupta, Smith & Diakiw, 2022*). A definitive diagnosis of ALA can be made by detecting motile trophozoites in liver pus, but this requires a relatively invasive procedure and is not a sensitive diagnostic method (*Janwan et al., 2022*). Some of the commercially available antigen detection rapid tests include *Entamoeba* Ag Rapid Test (CD Creative Diagnostics® Inc. Shirley, NY, USA), *Entamoeba* Rapid Test Kit (RapidFor™, Vitrosens Biyoteknoloji, Istanbul, Türkiye), and CerTest *Entamoeba* one step card test (Certest Biotec S.L., Zaragoza, Spain). These kits detect stool samples for *Entamoeba* infection in the human gastrointestinal tract. However, most patients with ALA have no history of gastrointestinal symptoms, and stool examination is often negative for cysts or trophozoites at the time of diagnosis (*Shirley et al., 2018*; *Gupta, Smith & Diakiw, 2022*). Molecular techniques are considered the new diagnostic standard due to their ability to directly detect *E. histolytica* DNA in liver aspirate samples, and show impressive sensitivity and specificity (*Portunato et al., 2024*). Ultrasonography, computed tomography (CT) and magnetic resonance imaging (MRI) are all highly sensitive for detecting liver abscesses (*Saidin, Othman & Noordin, 2019*; *Iritani et al., 2021*; *Gupta, Smith & Diakiw, 2022*). However, it is necessary to distinguish ALA from other pyogenic abscesses or necrotic tumors (*Wong et al., 2017*; *Saidin, Othman & Noordin, 2019*). Detection of antibodies against *E. histolytica* has been a useful tool for diagnosis of ALA, combined with clinical manifestations and imaging findings. Several assays to detect anti-amebic antibodies have been developed (*Dhanalakshmi, Meenachi & Parija, 2016*; *Wong et al., 2017*; *Beyls et al., 2018*; *Tachibana et al., 2018*; *Janwan et al., 2022*). Recently, enzyme-linked immunosorbent assay (ELISA), the most common technique used to investigate amebiasis, has been re-evaluated to test the diagnostic performance of the immunoglobulin G (IgG) subclasses using crude antigen derived from *E. histolytica* strain HK9. That study demonstrated the potential for IgG ELISA to diagnose extraintestinal amebiasis (*Janwan et al., 2022*). The TechLab *E. histolytica* II™ test (TECHLAB®, Inc., Blacksburg, VA,

USA), an enzyme immunoassay for the detection of circulating *E. histolytica* Gal/GalNAc lectin has been developed for stool samples and utilized in serum samples in Dhaka, Bangladesh (*Haque et al., 2000*). However, this conventional assay is rather complicated and time-consuming, requiring costly reagents and sophisticated equipment. Delayed diagnosis can lead to severe morbidity, hence rapid, simple and low-cost diagnosis of ALA is vital to permit prompt, appropriate treatment. Here, we report the first development of an immunochromatographic test (ICT) that can use whole blood as an alternative to serum-based ELISA and compared its diagnostic utility with that of serum-based ELISA.

## MATERIALS & METHODS

This study was conducted according to the guidelines of the Declaration of Helsinki and approved by the Human Research Ethics Committee of Khon Kaen University (HE664044, approved 30 November 2023).

### Antigen preparation

A soluble extract of *E. histolytica* was prepared as previously described (*Janwan et al., 2022*). The *E. histolytica* HK-9 (*Eh* HK9) strain (axenized trophozoites) was obtained from Dr. Nimit Morakote, Department of Parasitology, Faculty of Medicine, Chiang Mai University and maintained in laboratory axenic culture. Briefly, the *Eh* HK9 was cultivated in *Diamond, Harlow & Cunnick (1978)*'s medium following a previously published protocol. After 48–72 h culture, the trophozoite stages were collected and washed with normal saline solution (NSS) by centrifugation at $120\times g$ at room temperature for 5 min. Approximately $10^7$ cells/mL were obtained by resuspending the cell sediment in an adequate amount of NSS. The sediment was sonicated with an ultrasonic disintegrator and centrifuged at $10,000\times g$ for 30 min at 4 °C. The supernatant was dialyzed against distilled water containing protease inhibitors (Roche Applied Science, Basel, Switzerland). The dialyzed protein concentration was determined using the method described previously (*Lowry et al., 1951*). The supernatant was aliquoted and stored at −70 °C until use as the antigen.

### Immunochromatographic device preparation

The ICT device for ALA (named Amebiasis blood rapid test kit: Amebiasis-ICT) was developed using the *Eh* HK9 antigen. A schematic illustration of the Amebiasis-ICT is shown in Fig. 1. The assembly process was performed as reported previously (*Rodpai et al., 2021*) with some modifications. A nitrocellulose membrane was dispensed with 1 mg/mL of the *Eh* HK9 antigen and 1 mg/mL of goat anti-mouse IgG (Lampire Biological Laboratories, Pipersville, PA, USA) at the test line (T) and the control line (C), respectively, at a rate spray of 1 μL/cm using an XYZ Dispensing Platform (Bio-Dot, Irvine, CA, USA). To prepare the colloidal gold-conjugated pad, 40 nm colloidal gold particles were conjugated with 6 μg/mL of mouse anti-human IgG (Arista Biologicals Inc., Allentown, PA, USA) and sprayed onto a glass microfiber filter (Whatman Schleicher & Schuell, Dassel, Germany). For the two types of pads, the sample pad (Millipore C048, Millipore, Burlington, MA, USA) and absorbent pad (Whatman#470, Merck KGaA, Darmstadt, Germany) were dissolved in a running buffer before use. All of the above materials were laminated onto a plastic backing

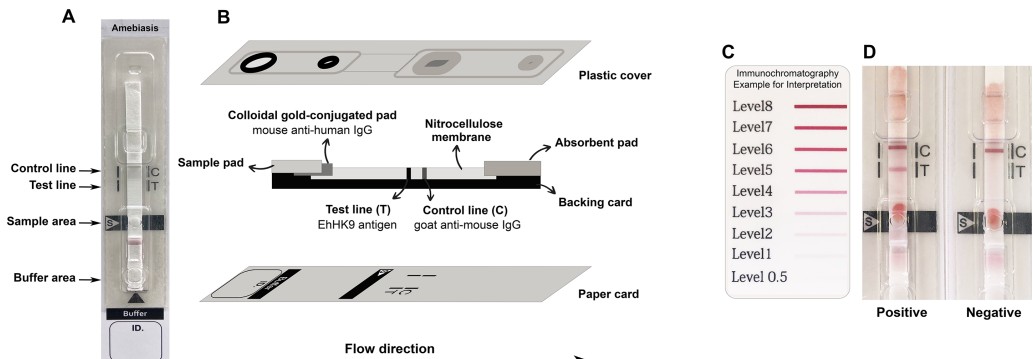

**Figure 1** **Schematic illustration of the Amebiasis ICT.** (A) and (B) are top and lateral views, respectively. (C) Color card for interpretation. The cut-off level is more than level 0.5. (D) Representative images of positive and negative results.

card, which was cut into strips five mm in width using a guillotine cutter (BioDot, Irvine, CA, USA). The strip was pasted onto a paper card, covered with a plastic housing (Fig. 1B) (Adtec Inc., Oita, Tokyo, Japan), then sealed with a desiccant in an aluminum foil bag, and stored at room temperature.

## Clinical samples and simulated whole-blood sample preparation

A total of 253 leftover serum samples (Table 1 and Table S1), which had been stored at the Frozen Serum Bank (Mekong Health Science Research Institute Biobank project), Faculty of Medicine, Khon Kaen University, Thailand, and Department of Parasitology, National Institute of Infectious Diseases, Tokyo, Japan, were used for evaluation of both ICT and ELISA. The serum samples were divided into three groups as follows: (i) ALA ($n = 13$), (ii) other confirmed parasitic or viral infections ($n = 200$), and (iii) negative controls ($n = 40$). Diagnosis of the ALA cases was based on the clinical presentations of amebic liver abscess, ultrasound and computerized tomography imaging findings, and serological confirmation tests against *E. histolytica* antigen, such as ELISA, immunofluorescence test, indirect hemagglutination assays, Ouchterlony's immunodiffusion test, and complement fixation test (*Janwan et al., 2022*). The parasitic infections had been confirmed by parasitological, serological, and/or histopathological examinations (*Elkins, Haswell-Elkins & Anderson, 1986*; *Intapan et al., 2008*; *Boonyasiri et al., 2014*). Viral hepatitis was diagnosed using the electrochemiluminescence immunoassay in the Cobas 8000 modular analyzer (Roche Diagnostics, Tokyo, Japan). The negative control group (iii) was proven to be free of parasites by the fecal formalin-ethyl acetate concentration method (*Elkins, Haswell-Elkins & Anderson, 1986*).

An equal volume of each ALA (group (i), $n = 10$) and negative control sera (group (iii), $n = 10$) were pooled to obtain the positive and negative reference sera, respectively. These reference sera were further used as control sera for determination of within-day and between-day precision in the ICT kits and ELISA plates. There were no variations within-day or between-day in the ELISA or ICT measurements caused by technical errors, fluctuations in environmental conditions, reagents, or similar factors.

**Table 1 Comparison of the diagnostic results using the Amebiasis-ICT and ELISA.**

| Types of clinical samples | Amebiasis-ICT | | ELISA | |
|---|---|---|---|---|
| | No. positive/ total | Intensities (Level)[a] | No. positive/ total | OD values[b] |
| Amebic liver abscess (ALA) | 13/13 | 1–5 | 13/13 | 0.433–1.899 |
| Giardiasis (Gl) | 1/14 | 0–1 | 0/14 | 0.072–0.263 |
| Blastocystosis (Bh) | 0/15 | 0 | 1/15 | 0.009–0.724 |
| Toxocariasis (Toxo) | 0/1 | 0 | 0/1 | 0.173 |
| Schistosomiasis mekongi (Smk) | 2/10 | 0–3 | 3/10 | 0.140–0.450 |
| Clonorchiasis (Cs) | 1/10 | 0–1 | 1/10 | 0.079–0.488 |
| Opisthorchiasis viverrini (Ov) | 0/10 | 0 | 0/10 | 0.038–0.255 |
| Paragonimiasis heterotremus (Ph) | 0/10 | 0 | 0/10 | 0.101–0.296 |
| Fascioliasis (Fg) | 1/10 | 0–1 | 2/10 | 0.100–0.601 |
| Taeniasis saginata | 0/9 | 0 | 0/9 | 0.056–0.257 |
| Sparganosis (Se) | 0/5 | 0 | 0/5 | 0.048–0.166 |
| Cysticercosis (Cc) | 1/10 | 0–1 | 1/10 | 0.027–0.412 |
| Hookworm infection (Hw) | 1/10 | 0–1 | 1/10 | 0.097–0.449 |
| Trichinellosis (Ts) | 0/10 | 0 | 0/10 | 0.065–0.182 |
| Capillariasis (Cp) | 0/10 | 0–0.5 | 1/10 | 0.053–0.524 |
| Gnathostomiasis (Gs) | 0/10 | 0 | 0/10 | 0.073–0.294 |
| Ascariasis (Al) | 0/10 | 0 | 0/10 | 0.064–0.157 |
| Strongyloidiasis (Ss) | 0/10 | 0 | 0/10 | 0.038–0.270 |
| Angiostrongyliasis (Ac) | 0/10 | 0 | 0/10 | 0.058–0.188 |
| Trichuriasis (Tt) | 0/10 | 0 | 0/10 | 0.034–0.255 |
| Hepatitis virus infection | 0/16 | 0–0.5 | 0/16 | 0.075–0.293 |
| Negative control (HC) | 0/40 | 0 | 0/40 | 0.048–0.275 |

**Notes.**

[a] Band intensity at the T line was evaluated according to the reference color card (>0.5 as the cut-off level).

[b] OD values were judged as positive if they exceeded the mean plus three times the standard deviation (>0.332) of the values obtained from 40 negative controls.

Simulated whole-blood samples (WBSs) were prepared to determine the diagnostic values of the ICT tool. Ethylenediaminetetraacetic acid anti-coagulated whole blood was obtained from a healthy volunteer whose stool sample revealed no intestinal parasitic infection. Half a milliliter of this was centrifuged at $12,000 \times g$ for 10 min at 4 °C and plasma was removed. The packed red blood cells (RBCs) were washed thrice with phosphate-buffered saline (PBS), pH 7.4, using centrifugation at $12,000 \times g$ for 10 min at 4 °C. The packed RBCs were re-suspended in equal volumes of PBS to the original volume of whole blood. The suspended RBC was divided into 20 μL-aliquots, which were then centrifuged at $12,000 \times g$ for 10 min at 4 °C. Thirteen microliters of the supernatant was discarded and the remaining 7 μL packed RBCs were utilized to generate simulated WBSs by adding 13 μL of the relevant serum sample.

## Immunochromatographic analysis of samples

To test the ICT device, each simulated WBS sample was diluted with sample buffer to 1:10 dilution. Then added 5 μL of diluted sample onto a sample well and 60 μL of running

buffer onto the buffer well. The result was visually interpreted at 15 min after the addition of the running buffer. The presence of the bands at both the "T" and "C" lines implies a positive result. The "T" band intensity can be evaluated visually according to the color card (Fig. 1C, >0.5 as the cut-off level).

## Enzyme-linked immunosorbent assay

ELISA was performed as reported previously (*Janwan et al., 2022*) with slight modification. Briefly, flat bottom plates were coated with 100 µL of the *Eh* HK9 antigen (2.5 µg protein/mL) in 0.1 M carbonate buffer, pH 9.6 for overnight at 4 °C. Phosphate buffer saline (0.1 M), pH 7.4 containing 0.05% Tween 20 (PBST) was used as a diluent and washing. The coated plates were washed five times with PBST, and blocked in PBST with 3% bovine serum albumin (BSA). To probe the primary antibody, a 100 µL of 1:800-diluted human serum sample in PBST with 1% BSA was added. After washing, the secondary antibody was probed by adding 100 µL of 1:100,000-diluted horseradish peroxidase-conjugated goat anti-human IgG (Invitrogen Corporation, Waltham, MA, USA), and the plates were incubated for 1 h at 37 °C. Following washing, 100 µL of *o*-phenylene diamine hydrochloride was added. Finally, the reaction was terminated by adding 50 µL of 8N sulfuric acid. Optical density (OD) was read at 492 nm and the cut-off value was set at the mean plus 3 standard deviations of the OD value from 40 negative controls (mean+3SD; $0.143 + 3(0.063) = 0.332$). OD values below this cut-off were scored as negative.

## Statistical analysis

The diagnostic values in the Amebiasis-ICT and ELISA were calculated as reported previously (*Galen, 1980*). Sensitivity, specificity and cross-reactivity of Amebiasis-ICT were statistically compared with those of the ELISA using McNemar's test. The degree of agreement between ELISA and the ICT results was calculated using Cohen's kappa coefficient. Stata version 10.1 was used to perform the analysis.

## RESULTS

The ICT and ELISA results are summarized in Table 1. Neither Amebiasis-ICT nor ELISA showed positive results in the 40 negative controls. By contrast, all serum samples from the ALA cases were positive. Cross-reactions in the Amebiasis-ICT test were revealed in seven cases (Table 1): giardiasis, *Schistosomiasis mekongi*, clonorchiasis, fascioliasis, cysticercosis, and hookworm infections. In ELISA, cross-reactions were also observed in blastocystosis, *Schistosomiasis mekongi*, clonorchiasis, fascioliasis, cysticercosis, and hookworm infections. The diagnostic sensitivity and specificity values in the Amebiasis-ICT test were computed as 100% and 97.1%, respectively, and those of the ELISA were 100% and 95.8%, respectively (Table 2). There were no significant differences in sensitivity and specificity between the two tests (*p*-value >0.05), with a concordance of 96.4% represented by Cohen's kappa agreement of 0.771 (*p*-value <0.001; Table 2), which indicated substantial agreement.

**Table 2 Diagnostic values of Amebiasis-ICT and the ELISA for diagnosis of human amebic liver abscess.**

| Diagnostic values | Amebiasis-ICT[#] (95% Confidence interval) | ELISA[#] (95% Confidence interval) |
|---|---|---|
| Sensitivity (%) | 100 (75.3 to 100) | 100 (75.3 to 100) |
| Specificity (%) | 97.1 (94.1 to 98.8) | 95.8 (92.5 to 98.0) |
| Positive predictive value (%) | 65.0 (40.8 to 84.6) | 56.5 (34.5 to 76.8) |
| Negative predictive value (%) | 100 (98.4 to 100) | 100 (98.4 to 100) |
| Positive likelihood ratio | 34.3 (16.5 to 71.1) | 24.0 (13.1 to 44.0) |
| Negative likelihood ratio | 0 | 0 |
| Accuracy (%) | 97.2 (94.4 to 98.9) | 96.0 (92.9 to 98.1) |

**Notes.**

[#]There was no significant difference between Amebiasis-ICT and ELISA (Exact McNemar's test; $p$-value $= 0.508$), with Cohen's kappa agreement 96.44% ($\kappa$-value $= 0.771$, $p$-value $< 0.001$).

# DISCUSSION

To our knowledge, no specific serodiagnostic assays for ALA can make use of whole-blood samples: all have used serum samples. In this study, a new IgG antibody detection-based ICT using simulated whole-blood samples was developed as a point-of-care test (POCT) for serodiagnosis of ALA. Its diagnostic performance was compared with an ELISA using serum samples. Both tests had a diagnostic sensitivity of 100%. The specificity of the ICT test (97.1%) was slightly higher than that of ELISA (95.8%). These results corresponded well with diagnostic sensitivity for antibody detection by an ICT kit using fluorescent silica nanoparticles (*Tachibana et al., 2018*) and ELISA targeting the IgG antibody (*Janwan et al., 2022*) and was higher than some previous ELISA reports (69.0–99.7%) (*Dhanalakshmi, Meenachi & Parija, 2016*; *Kannathasan et al., 2017*; *Beyls et al., 2018*). Meanwhile, the sensitivity of the *E. histolytica* II™ antigen (TechLab Inc., Blacksburg, VA, USA), an ELISA-based technique for antigen detection, showed 96% in ALA sera (*Haque et al., 2000*). The diagnostic specificity of the present ICT test (97.1%) was close to that of the ICT kit using fluorescent silica nanoparticles (97.6%) (*Tachibana et al., 2018*) and ELISA targeting the IgG antibody (90.0–99.7%) (*Dhanalakshmi, Meenachi & Parija, 2016*; *Kannathasan et al., 2017*; *Janwan et al., 2022*; *Beyls et al., 2018*). The *E. histolytica* quik chek™ (TechLab Inc., Blacksburg, VA, USA) has been reported as a POCT kit that showed high sensitivity and specificity (100%) for determination the presence of *E. histolytica* gastrointestinal infection in fecal samples of patients with diarrhea or dysentery (*Korpe et al., 2012*). Additionally, the XEh Rapid® , an IgG4-based rapid dipstick test showed diagnostic specificity (97–100%) and sensitivity was variable from high to low (38–94%) in multi-laboratory evaluation (*Noordin et al., 2020*). The different values of diagnostic sensitivity and specificity might depend on differences in technique used (ELISA *vs* ICT), optimum conditions (such as buffer systems, types and concentration of antigens, kinds of membranes, and type of plastic covers), type of evaluated samples (whole-blood or serum samples) and differences in the individual samples. Our ICT has two benefits over commercially available tools: (1) The quality of the kit was assertively shown, when kept in an aluminum foil bag with silica gel desiccant for 12 months at room temperature (25 °C) and for 18 months at 4 °C,

during which it maintained its activity and stability in band intensity when testing with the positive and negative reference sera. (2) The diagnostic values of the assay can be optimized for each endemic zone by use of local positive and negative control samples in each zone. Moreover, the ICT can provide results much more quickly, typically within 15 min, making it suitable for urgent diagnostic needs. While the ELISA test requires approximately 3 h to complete, starting from the addition of serum.

However, our Amebiasis-ICT exhibited some cross-reactions in giardiasis (1/14; 7.1%), *Schistosomiasis mekongi* (2/10; 20%), clonorchiasis (1/10; 10%), fascioliasis (1/10; 10%), cysticercosis 1/10; 10%), and hookworm infections (1/10; 10%). Generally, cross-reactions with other parasitosis samples do not cause a major question for clinical diagnosis. Clinical symptoms of hookworm infection and giardiasis differ from those of ALA and normally reveal a diagnostic stage in fecal samples. Even though clonorchiasis, cysticercosis, fascioliasis, and schistosomiasis can present clinical features and liver inflammation/abscess cognate to ALA, they also reveal with the different of the history of infection, stool determination findings, peripheral blood eosinophilia, medical imaging abnormalities of the liver, and/or medication responses.

Cross-reactions might be due to collection of serum samples from donors who may have had asymptomatic infections with other parasites. Co-infections with *E. histolytica* and *Giardia* spp. in particular might make it more difficult to interpret test findings because antibodies from previous infections may react with the antigens used in tests, potentially producing false positives. Particularly in areas where both are prevalent, the frequency of these co-infections shows a need for careful diagnostic techniques to accurately differentiate between these two protozoans. It is also possible that cross-reactivities may be a consequence of antigenic epitopes shared with some other parasites. Differential diagnosis should include clinical presentation as well as consideration of history of travel to endemic areas and radiological findings (CT and MRI) and other laboratory blood findings. Nevertheless, clinicians and laboratory technologists should be aware that some cross-reactions with giardiasis, *Schistosomiasis mekongi*, clonorchiasis, fascioliasis, cysticercosis, and hookworm infection sera may occur when using this kit.

The limitations of this antibody based-ICT test are (1) the test cannot distinguish past from current infection, and does not test with other amebiasis. (2) Clinicians and technicians should be aware, in amebiasis diagnosing using this POCT kit, that the test has only been evaluated in a laboratory setting using a defined set of simulated whole-blood samples and a small number of tested samples. The performance of the kit still needs to be evaluated with real whole-blood samples. (3) Diagnostic values may vary according to the samples with which the kit is used. (4) In the present study, crude soluble extract of antigen was obtained from *E. histolytica* trophozoites. To improve this tool, highly sensitive and specific antigens produced by recombinant technology should be applied. This would guarantee a stable antigen supply and simplify quality control.

## CONCLUSIONS

We reported a new POCT kit for detection of anti-amebic IgG antibody using simulated whole-blood samples for serodiagnosis of ALA. The ICT platform is simple to use with

a short turnaround time. The diagnostic values of the Amebiasis-ICT and ELISA did not differ significantly ($p$-value > 0.05), representing that the ICT tool can be substituted for the antibody detection-based ELISA in human sera for serodiagnosis of ALA. More advantages of this POCT kit are that the ICT tool does not require skilled personnel to use, and it might be possible to apply with fingerstick blood samples, thereby eliminating the need to use venous blood and separate serum. The kit is not only easy to support clinical diagnosis at the bedside but also for diagnosing infected cases in field-based work in endemic and distant regions where complicated equipment may not be available.

## ACKNOWLEDGEMENTS

We thank Prof. David Blair for the English editing of this manuscript.

### Funding

This project was funded by grants from the National Research Council of Thailand (NRCT): High-Potential Research Team Grant Program (Contract no. N42A670561 to Wanchai Maleewong) and the Research Program from Research and Graduate studies, Khon Kaen University (KKU) (grant no. RP66-7-001 to Wanchai Maleewong). The contents of this report are solely the responsibility of the authors and do not necessarily represent the official views of any grant-awarding body. The funders had no role in study design, data collection and analysis, decision to publish, or preparation of the manuscript.

### Grant Disclosures

The following grant information was disclosed by the authors:
National Research Council of Thailand: N42A670561.
Research Program from Research and Graduate Studies, Khon Kaen University (KKU): RP66-7-001.

### Competing Interests

The authors declare there are no competing interests.

### Author Contributions

- Rutchanee Rodpai conceived and designed the experiments, performed the experiments, analyzed the data, prepared figures and/or tables, authored or reviewed drafts of the article, and approved the final draft.
- Penchom Janwan conceived and designed the experiments, performed the experiments, analyzed the data, authored or reviewed drafts of the article, and approved the final draft.
- Lakkhana Sadaow conceived and designed the experiments, performed the experiments, analyzed the data, authored or reviewed drafts of the article, and approved the final draft.
- Patcharaporn Boonroumkaew performed the experiments, authored or reviewed drafts of the article, and approved the final draft.

- Oranuch Sanpool conceived and designed the experiments, authored or reviewed drafts of the article, and approved the final draft.
- Tongjit Thanchomnang conceived and designed the experiments, authored or reviewed drafts of the article, and approved the final draft.
- Hiroshi Yamasaki conceived and designed the experiments, authored or reviewed drafts of the article, and approved the final draft.
- Toshihiro Mita conceived and designed the experiments, authored or reviewed drafts of the article, and approved the final draft.
- Pewpan M. Intapan conceived and designed the experiments, authored or reviewed drafts of the article, and approved the final draft.
- Wanchai Maleewong conceived and designed the experiments, authored or reviewed drafts of the article, and approved the final draft.

### Ethics

The following information was supplied relating to ethical approvals (i.e., approving body and any reference numbers):

This study was conducted according to the guidelines of the Declaration of Helsinki and approved by the Human Research Ethics Committee of Khon Kaen University (HE664044, approved 30 November 2023).

### Data Availability

The individual measurements are available in the Supplemental Files.

### Supplemental Information

Supplemental information for this article can be found online at http://dx.doi.org/10.7717/peerj.19181#supplemental-information.

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
