# Peer review of "A proof-of-concept point-of-care test for the serodiagnosis of human amebic liver abscess"

_PeerJ, doi:10.7717/peerj.19181_

## Round 0.1 · original submission · Major Revisions

Dear Author:

The submitted manuscript is relevant. The change to the methodological article has a better scientific sound. Nevertheless, several changes are needed to improve the content and validation of the results.

Also, the positive and negative controls needed to be clarified. The same is true for cross-reactivity assays for the PoC samples.

It is relevant that you declare the limitations and perspective of the manuscript only based on the results.

Include a point-by-point response to reviewers.

Reviewer 1 ·

Basic reporting

I am not native speaker; therefore I am not considered myself an authority to correct the language.
The introduction and the backgrounds are short, but they contain the necessary information to introduce the need and description of the study.
It accomplished with the Peer J. standards
It only contains one figure. It is well described, and I consider is enough to describe the design of the lateral flow system.

The authors dont mention clearly the source of the EhHK9 strain

Experimental design

This work covers topics related to health science and I consider it covers the scope of the journal.
The authors highlight the relevance of the research as they describe the need of having the development of the CT platform necessary for the diagnostic of ALA, which is a neglected disease in undeveloped countries.
The author presents the complete results without hiding any of them that can compromise the final development and commercialization of the platform.
The methods describe in detail the components of the CT platform, with the specification to obtain replicates.

Validity of the findings

The study present original results, it is not replication or any other study.
The authors present the data that support the study. They report that the total number of samples analyzed are 253, out of them, 13 are positive to ALA. I consider this number is mall and from this mall number they don’t report false positives, and 40 negative samples that also don’t present false negatives.

When they present the false positives using clinical samples from other diseases, in most of the cases they are positive for ELISA and ICT platform. These results imply that the results are inherent to the ICT, as the cross reactivity is the most common problem in this type of samples.
Based on the above observations, I consider that the conclusions where they claim that CT platform is the sero diagnostics of ALA are risky. Even if it is correct for a sero diagnostics of amebiasis, it is necessary to contrast with the clinic information of the patient to discriminate amebiasis vs. ALA.

Additional comments

Based on the sample size and the results obtained, I suggest changing the title, even when it is innovative, I consider is still a proof of concept of the sero diagnostics platform.
By definition the Amebic liver abscess (ALA) is a severe extraintestinal manifestation caused when pathogenic trophozoites of E. histolytica disseminate to the liver, at this point the CT platform does not discriminate if the antibodies produced in the patients correspond to this type of infection or they correspond to another clinical manifestations of the disease. It is is necessary to include the clinical studies and the image analyses.

Reviewer 2 ·

Basic reporting

no comment

Experimental design

no comment

Validity of the findings

no comment

Additional comments

The authors present a study of a new point-of-care (POC) serological test for diagnosing amoebic liver abscess (ALA) in areas where Entamoeba histolytica is endemic. I appreciate the authors' efforts to address the limitations of traditional diagnostic methods and their ambition to create a more accessible alternative. Below, I offer a few comments and suggestions to improve the clarity of the manuscript.

Main observations and comments,
1/ While the authors aim to evaluate a serological test for ALA in an endemic region, a more precise approach would be to stratify participants according to their serological status confirmed by a reference ELISA test. This would enable a clearer correlation between the new POC test and the ELISA test, particularly within defined groups (ALA and non-ALA).
2/ In addition, the study could benefit from an evaluation using spiked serum samples as controls, especially as this is a methodological paper with limited results. Greater attention to method validation would strengthen the study.
3/ With regard to statistical analysis, given the small sample size of ALA-positive patients, focusing on sensitivity, positive predictive value (PPV) and negative predictive value (NPV) may not be statistically robust. Instead, focusing on kappa agreement may provide a clearer representation of test accuracy. The intensity levels with ICT are briefly mentioned but would benefit from a direct comparison with ELISA OD ranges. In addition, did the authors consider serial dilutions with a known titer to evaluate intensity levels systematically?

Minor comments,
The introduction is somewhat verbose and could be simplified by removing non-essential details. The current introductory angle seems generic. Given that the proposed assay may not be ideal for diagnosing ALA in endemic regions, reworking the introduction to align it more specifically with the study objectives would improve relevance. For example:
- Lines 58-61: The description of non-endemic areas and risk factors might be condensed or omitted.
- Lines 65-67: Details on ALA reports in travelers could be shortened, as it may be less relevant to the study’s primary focus.
Lines 67-69: The gold standard for ALA diagnosis is not detecting trophozoites in liver pus, as less than 10% of cases yield positive results due to ALA pathophysiology. Instead, PCR in liver pus with 100% sensitivity should be considered the gold standard.
Lines 83-88: While the study describes the ELISA test as time-consuming, the POC method described seems equally time-consuming in terms of preparation. These details would be better suited to the “Discussion” section.
Line 103: Replace “proteinase inhibitors” with “protease inhibitors.”
About sample selection: For ALA cases diagnosed based on serology and imaging in endemic areas, the possibility of bias should be acknowledged. A PCR-based diagnosis or collaboration with a center performing PCR would mitigate this issue. In addition, other parasitic/viral infections were confirmed through fecal examinations, not serology. Using samples with known positive serologies would provide a stronger comparison and validate the test’s cross-reactivity more accurately.
Line 214: The authors mention the possibility of using whole blood samples, but their analytical assessment is based on serum samples. This discrepancy should be corrected and further clarification provided on whole blood analysis.
Line 220: The ICT cross-reacted with Giardia infections, although this parasite does not typically induce circulating antibodies due to its strict intraluminal localization. The authors should discuss this anomaly and provide insights into the potential causes of these cross-reactions.
Line 237: The authors might consider promoting the ICT as an antibody detection test rather than a direct ALA diagnostic tool. The correlation between ELISA titers and ALA diagnosis is well-established; it would be valuable to explore whether a similar correlation exists with the POC test described.

Some grammatical issues,
Line 51: "is one of the important neglected tropical diseases" would be more impactful as "is one of the most important neglected tropical diseases."
Line 55: Prefer “low-income countries” over “developing countries” for contemporary sensitivity and accuracy.
Line 86: Replace “targeted treatment” with “appropriate treatment” for clarity.

---

## Round 0.2 · Major Revisions

Dear Authors:
It has not been easy to find reviewers for your manuscript. I now have enough recommendations that must be addressed before publishing the manuscript. Promptly provide a point-by-point response to the reviewer 3. After this, I will quickly follow up on the final decision for this manuscript.

Reviewer 2 ·

Basic reporting

no comment

Experimental design

no comment

Validity of the findings

no comment

Additional comments

no comment

Reviewer 3 ·

Basic reporting

See #4, additonal comments

Experimental design

See #4, additonal comments

Validity of the findings

See #4, additonal comments

Additional comments

In this manuscript, the authors report a point-of-care test for Entamoeba histolytica infection in blood. They developed an immunochromatographic test using antigen to detect the Entamoeba antibodies. Clinical serum samples from infected and uninfected patients were diluted with blood cells to create simulated whole blood samples. These were evaluted by the immunochromatographic test and ELISA. Overall, the authors provided a fairly preliminary report.

It's important to note that many commercially available immunochromatographic devices are available for Entamoeba detection: Entamoeba Ag Rapid Test (Creative Diagnostics Inc.), E. HISTOLYTICA II (TechLab Inc.), Entamoeba Rapid Test Kit (Vitrosens Biyoteknoloji), CerTest Entamoeba one step card test (Certest Biotec S.L.), etc. The authors need to acknowledge this in the introduction and discussion sections. The authors need to differentiate their study with these commercial devices.

In the discussion, the authors should compare their results to those obtained by other immunochromatographic tests for Entamoeba. For example, doi: 10.4269/ajtmh.20-0348, 10.4269/ajtmh.2012.11-0661, 10.1128/jcm.38.9.3235-3239.2000, etc.

The authors took clinical samples and spiked them with red blood cells to simulate whole blood, which was analyzed by the device. Commercially available immunochromatographic devices for Entamoeba analyze fecal/stool samples, which are far less invasive than obtaining whole blood. This is a notable omission by the authors that undermines the rationale for this study.

The authors mention that some false positives could be due to previous Entamoeba histolytica infections. Therefore, authors need to change the title and text throughout to say that this is a test for Entamoeba histolytica infection, not ALA since a limited number of positive samples and unorthodox way of using simulated blood samples does not actually demonstrate true diagostic tool for ALA.

The methods for device preparation and sample analysis by the device should be separatated to make the methods follow a more logical order. Please move lines 142-158 to follow "Antigen Preparation"; then create a new section heading for lines 159-164 called "Immunochromatographic Analysis of Samples" (or something similar).

---

## Round 0.3 · accepted · Accept

Considering the modification and suggestion for reviewer 3, your manuscript is accepted.

Reviewer 3 ·

Basic reporting

na

Experimental design

na

Validity of the findings

na

Additional comments

na